# The Impact of Self-Transcendence on Anxiety Among Chinese College Students: The Moderating Roles of Self-Enhancement and Dominant Self-Construal

**DOI:** 10.3390/bs14111105

**Published:** 2024-11-17

**Authors:** Hong Wang, Tong Yue, Huajun Luo

**Affiliations:** 1Key Laboratory of Applied Psychology, Chongqing Normal University, Chongqing 401331, China; 20131691@cqnu.edu.cn; 2School of Teacher Education, Chongqing Normal University, Chongqing 401331, China; 3College of Educational Science, Chongqing Normal University, Chongqing 401331, China; lhj1061267663@163.com; 4Research Center for Psychology and Social Development, Southwest University, Chongqing 400715, China; 5School of Psychology, Southwest University, Chongqing 400715, China

**Keywords:** self-transcendence, self-enhancement, self-construal, anxiety, mental health, moderated moderation

## Abstract

This study explores the role of self-transcendence values in reducing anxiety among Chinese undergraduates, with a particular focus on how self-enhancement values and self-construal styles moderate this relationship. A total of 567 undergraduate students were assessed using the Portrait Values Questionnaire (PVQ-21), the Trait Anxiety Inventory (T-AI), and the Self-Construal Scale (SCS). Hierarchical regression and moderated moderation analyses were conducted to test the interactions between the variables. The findings suggest that self-enhancement moderated the association between self-transcendence and anxiety, particularly in individuals with low self-enhancement. Furthermore, the moderating effect of self-enhancement was influenced by dominant self-construal. These findings have practical implications for integrating value-based interventions in educational and public health strategies aimed at improving mental well-being among university students.

## 1. Introduction

### 1.1. The Effect of Self-Transcendence Values on Anxiety

Values play a crucial role in shaping individual behavior and influencing mental well-being. Schwartz’s theory of basic human values identifies self-transcendence and self-enhancement as two value dimensions with significant implications for psychological health across different cultural contexts [1]. Research from various countries has shown that these values can affect mental health in both individualistic and collectivist societies, suggesting that the relationship between values and anxiety is a global phenomenon. Further, prolonged exposure to chronic anxiety has been consistently linked with negative mental health outcomes. Studies indicate that chronic anxiety not only affects present well-being but also escalates the risk of comorbid psychiatric and somatic disorders. For instance, Kessler et al. [2] found that anxiety disorders frequently co-occur with depression and substance abuse, and Sareen et al. [3] demonstrated that individuals with high trait anxiety are more likely to experience a range of mental health challenges if not addressed. In the context of public health, understanding how these values interact to influence anxiety in various populations, including Chinese undergraduates, can guide the development of targeted mental health interventions that are culturally sensitive and effective in promoting well-being.

Schwartz’s theory of human core values is one of the most widely used models in psychology. It identifies ten core values that are categorized into four higher-order dimensions: self-transcendence, self-enhancement, conservation, and openness to change [1,4]. This theory has been applied globally, from Western individualistic cultures to Eastern collectivist societies, highlighting its broad applicability. To further differentiate these values, Schwartz identifies two structuring principles [5]. The first principle concerns the interests served by the achievement of a particular value. Personal-focus values emphasize self-interest, relative success, and power over others, whereas social-focus values emphasize concern for the well-being and interests of others. The second principle is particularly relevant to mental health interventions, as it relates to how values connect to anxiety. Anxiety-free values represent growth and self-expansion, leading to feelings of meaningfulness and satisfaction. Conversely, anxiety-avoidant values are driven by the need to protect oneself and avoid or regulate anxiety and threats, potentially leading to negative emotions such as depression and anxiety.

Self-transcendence values, which emphasize social well-being and personal growth, have been linked to lower anxiety and greater life satisfaction in various cultural contexts, making them particularly relevant for mental health promotion within educational settings globally [4]. In contrast, self-enhancement values, which prioritize personal achievement, have consistently been associated with higher levels of anxiety and stress across different populations. These value dynamics, when considered from a public health perspective, can provide actionable insights for designing interventions that reduce anxiety and enhance quality of life not only in Chinese students but also in diverse educational environments. Scholars have long been interested in exploring the relationship between self-transcendence values and mental health. For example, studies in both Russia and China have identified a negative association between self-transcendence and depression. Research from both collectivist cultures, such as China, and individualist cultures has shown that self-transcendence values are associated with higher levels of happiness and greater well-being [5,6]. A recent study by Xie et al. [7]. found that Chinese college students who valued self-transcendence experienced fewer mental health issues and higher life satisfaction. Similar patterns have been observed in other cultural settings, suggesting that individuals with socially focused values are more empathetic [8,9,10], more likely to have positive interpersonal relationships [11], receive higher levels of social support for coping with negative life events [12], and exhibit fewer mental health disorders, such as anxiety [6,13]. Thus, this cross-cultural evidence supports the first objective of this study, which is to verify whether self-transcendence values are effective in reducing anxiety across different cultural contexts [14].

### 1.2. Self-Enhancement Values as a Moderator

Schwartz’s [1] conceptual model of values suggests that individuals with stronger self-enhancement values are more driven and focus on achieving their self-interest by controlling other people and resources. Given the structural conflict between self-transcendence and self-enhancement values, their effects on mental health are expected to differ significantly. Self-enhancement values, which emphasize personal focus and anxiety avoidance, are fundamentally incompatible with self-transcendence values in terms of motivation. Furthermore, unlike self-transcendence, which promotes mental health, research across various populations has shown that self-enhancement values can have a detrimental impact on mental health. For instance, a study conducted in Germany found that achievement was positively related to depression in university students [15]. Similarly, studies in different cultural contexts have found that adolescents who emphasize self-improvement often become more self-focused and preoccupied with others’ opinions of them, making them more vulnerable to self-threatening situations and resulting in emotions such as anxiety, depression, and sadness [16,17]. Other studies have found that individuals who prioritize personally oriented values and are overly focused on their own interests and successes may experience greater stress [15] and be more likely to internalize issues [13].

Since these two value orientations are structurally in conflict and have opposing effects on mental health, it raises the question of whether they interfere with each other when individuals identify with both. This issue has been explored in several studies across different cultures. For instance, Burroughs and Rindfleisch [18] found that individuals with high levels of socially focused values (e.g., religious and family values) who also identified with personally focused values (e.g., materialism) experienced higher levels of depression, anxiety, and stress. In contrast, individuals with low levels of personally focused values did not exhibit this pattern. Similarly, Furchheim, Martin, and Morhart [19] found that subjects who scored high on both green and materialistic values reported higher levels of perceived stress, leading to a decrease in life satisfaction. These findings suggest that the simultaneous pursuit of conflicting values, across various cultural settings, can hinder goal-directed behavior and encourage the emergence of psychopathological symptoms [20].

Recent large-scale social changes in China have increasingly focused on the realization of personal goals. Schools and parents have begun to encourage students to develop independence, self-expression, and self-exploration. As a result, many Chinese college students pursue both self-transcendence and self-enhancement values simultaneously. However, similar patterns have been observed in other cultures experiencing rapid modernization, such as South Korea and Japan, where young adults are also encouraged to balance traditional collectivist values with modern individualistic pursuits. Despite this global trend, few studies have explored how the simultaneous pursuit of these two value types affects the mental health of college students. Previous research suggests that the pursuit of self-transcendence and self-enhancement values may interfere with each other. Building on this foundation, we aim to explore whether there is a boundary condition for the mitigating effect of self-transcendence values on anxiety. Therefore, we hypothesize in this study that self-enhancement values moderate the relationship between self-transcendence values and anxiety.

### 1.3. Dominant Self-Construal as a Second-Order Moderator

According to self-construal theory, a key concept in cultural psychology, dominant self-construal refers to the relative strength of independent versus interdependent self-construal in influencing behavior and cognition. At the individual level, both independent and interdependent self-construal can coexist, and it is the relative strength of each (i.e., dominant self-construal) that determines which form drives an individual’s behavior. Dominant self-construal refers to how individuals relate their self-concept to others—whether more independently or interdependently. Cross-cultural research has consistently found that individuals in Western cultures, such as the United States and Europe, are more likely to construe themselves independently, whereas those in Eastern cultures, such as China and Japan, are more likely to construe themselves interdependently with others [21].

Research has shown that value conflicts and confrontations often depend on how individuals define and construct their self-concept [22]. In general, the less importance a person places on the clarity and consistency of their self-concept, the more compatible their differing values will be, reducing the likelihood of value conflicts. This is because value conflicts can negatively impact an individual’s ability to perceive themselves clearly, thus threatening the integrity of their self-concept, which can lead to various psychological symptoms. The concept of Self-Concept Clarity (SCC) is more prominent in Western cultures, where internal consistency and stability are emphasized, and individuals are expected to maintain a clear sense of identity across different contexts. In contrast, individuals with interdependent dominant self-construal, common in Eastern cultures, often engage in dialectical thinking, making their self-concepts less consistent and thus potentially more tolerant of value conflicts [23]. Considering the interaction between self-transcendence and self-enhancement values, and the varying tolerance of dominant self-construals towards value conflicts, we hypothesize that, for individuals with an independent dominant self-construal, self-enhancement values may moderate the relationship between self-transcendence values and anxiety. However, for individuals with an interdependent dominant self-construal, the interaction between self-transcendence and self-enhancement values may not significantly impact anxiety.

### 1.4. The Present Study

Chinese university students live in an era where tradition and modernity coexist, enabling both interdependent and independent self-construal to shape their behavior. Similar dynamics can be observed in other rapidly modernizing societies, such as South Korea and Japan, where young adults are also balancing traditional values with modern individualistic aspirations [24]. This context provides a valuable opportunity to examine the relationship between values and mental health through the lens of self-construal differences. Based on theoretical considerations and findings from the literature, we developed a model (see Figure 1) to explore potential relationships among these variables. Our investigation focused on the following hypotheses:

**Hypothesis** **1.**
*Self-transcendence values are negatively associated with anxiety.*


**Hypothesis** **2.**
*Self-enhancement values moderate the relationship between self-transcendence values and anxiety.*


**Hypothesis** **3.**
*Self-construal moderates the role of self-enhancement in the relationship between self-transcendence values and anxiety.*


## 2. Methods

### 2.1. Design

This study employed a quantitative survey method with a cross-sectional design to explore the relationship between self-transcendence, self-enhancement, and anxiety among Chinese college students. The cross-sectional design was selected for its efficiency in capturing data at a single point in time, allowing for an analysis of correlations between variables in a large sample. According to Creswell and Creswell’s research framework [25], such a design is appropriate for identifying associations without establishing causality.

### 2.2. Setting

Data were collected in a classroom setting at urban universities in Chongqing, Shandong, and Tianjin, China, with the permission of the university administrations. The data collection period was from September to November 2023.

### 2.3. Participants

A total of 588 students from urban universities in Chongqing, Shandong, and Tianjin were selected using convenience sampling. After excluding 21 participants who did not complete the surveys, 567 participants (96.42%) remained, comprising 174 men and 393 women, aged 17 to 25 years (*M* = 19.88, *SD* = 1.41).

The choice of urban university students was intentional for two key reasons. First, urban areas in China have been experiencing rapid modernization, leading to unique pressures on students, such as high academic expectations and career demands. These pressures are particularly relevant when studying anxiety levels and value-based behavior in educational settings. Second, students in urban areas are more likely to be influenced by both traditional collectivist values and emerging modern individualistic values, making them an ideal group to investigate the interaction between self-transcendence and self-enhancement in relation to anxiety.

### 2.4. Outcome Measures

Portrait Values Questionnaire (PVQ-21): The Chinese version of the Portrait Values Questionnaire (PVQ-21) was utilized to assess the respondents’ personal values. The PVQ-21 has been widely used in cross-cultural studies to measure the importance of different values across populations [26], and its psychometric properties have been validated in previous research. In Chinese samples, the PVQ-21 has shown good internal consistency, with Cronbach’s alpha values ranging from 0.70 to 0.80 across various value dimensions. Additionally, confirmatory factor analyses have demonstrated good construct validity of the scale in both Chinese and international contexts. Each portrait in the questionnaire conveys a subject’s objectives, hopes, or desires, subtly highlighting the significance of a particular value. Respondents answered the question, “How much like you is this person?”, using a scale from 1 = very much like me to 6 = not at all like me.

Trait Anxiety Inventory (T-AI): To measure anxiety, we used the Trait Anxiety Inventory (T-AI), which was originally developed by Spielberger et al. in 1970 [27] and later adapted to Chinese in 1995 [28]. This questionnaire includes 40 self-report items, divided into two subscales: the trait anxiety subscale, which measures a generally stable proneness to anxiety, and the state anxiety subscale, which measures the transient emotional state at the present moment. Given that college students often face unique academic and social pressures, trait anxiety scores may be higher in this population as compared to the general adult population. To address potential sample bias, we conducted analyses to ensure that scores were reflective of clinical relevance, with a focus on identifying ‘anxiety-related problems’ as indicated by a score above the threshold commonly associated with elevated anxiety [29]. This ensures that the anxiety measured aligns with clinically significant levels rather than normative responses to environmental stress. The Chinese version was validated with a Cronbach’s alpha of 0.91 for the trait anxiety scale and 0.88 for the state anxiety scale; these values are similar to those of the original version. Each subscale consists of 20 items, scored on a 4-point Likert scale: 0 = almost never/not at all, 1 = sometimes/somewhat, 2 = often/moderately so, and 3 = almost always/very much so. Because the trait subscale assesses stable rather than situational anxiety levels, it is suitable for screening college students, military personnel, and other professionals for anxiety-related problems, aligning with the purpose of this study. Therefore, the Trait Anxiety Inventory (T-AI) was selected to assess the daily anxiety levels of college students, with higher scores indicating more severe anxiety. In this study, the Cronbach’s alpha for the T-AI was 0.85, indicating good internal consistency.

Self-Construal Scale (SCS): Self-construal was measured using the Self-Construal Scale (SCS), which distinguishes between interdependent and independent self-construal. This scale, developed by Singelis in 1994, consists of 24 items—12 items each for the interdependent and independent subscales [30]. Respondents answer items using a 7-point Likert scale ranging from 1 (strongly disagree) to 7 (strongly agree). The Chinese version of the SCS has shown acceptable psychometric properties, with Cronbach’s alpha values of 0.88 for interdependent self-construal and 0.75 for independent self-construal in previous studies [31]. The interdependent self-construal subscale measures how much individuals see themselves as connected with others, emphasizing relationships and social harmony, while the independent self-construal subscale assesses how individuals see themselves as autonomous and unique. This scale has been widely used in cross-cultural psychology to assess how individuals define their relationships with others and their sense of independence.

The selection of these specific scales is grounded in their strong theoretical relevance to the constructs under investigation. The PVQ-21, T-AI, and SCS are all widely recognized tools that have been validated in diverse cultural contexts, making them appropriate for the current study examining the relationship between values and anxiety among Chinese college students.

### 2.5. Bias

Potential sources of bias include self-reporting bias due to the questionnaire format, which may influence participants’ responses. To mitigate this, anonymity was ensured, and the voluntary nature of participation was emphasized. Additionally, convenience sampling may limit generalizability to broader populations beyond urban university students in China. To check for common method bias, Harman’s single-factor test was conducted, identifying 13 distinct factors with eigenvalues greater than 1. The highest factor accounted for 13.52% of the variance, below the critical threshold of 40%, suggesting that common method bias was not significant [32]. A single-factor Confirmatory Factor Analysis (CFA) further assessed common method bias, showing poor fit indices (χ^2^/df = 58.23, CFI = 0.21, TLI = 0.10, RMSEA = 0.27, SRMR = 0.13), indicating that common method bias did not compromise the validity of the statistical findings.

### 2.6. Sampling and Study Size

The study began with an initial sample of 588 students, yielding 567 valid responses after exclusions. This sample size was determined to be sufficient for detecting correlations among self-transcendence, self-enhancement, and anxiety. Urban university students were chosen due to their exposure to modernization pressures and a mix of collectivist and individualistic values, providing a relevant population for this research.

### 2.7. Statistical Methods

Data analysis was conducted using SPSS 25 software. An overall descriptive analysis was followed by correlation analysis to examine relationships between variables. Hierarchical regression analysis was employed to investigate the moderating influence of dominant self-construal and self-enhancement values. For two-way interactions, a three-step regression analysis was conducted, including Model 0 (control variables), Model 1-1 (self-transcendence values and self-enhancement values/dominant self-construal), and Model 1-2 (interaction term). Control variables, such as age and gender, were included in Model 0 to account for their potential confounding effects on anxiety.

A four-step regression analysis tested the three-way interaction among self-transcendence values, self-enhancement values, and dominant self-construal, with Model 0, Model 2-1 (self-transcendence values, self-enhancement values, and dominant self-construal), Model 2-2 (two-way interaction terms), and Model 2-3 (three-way interaction term). PROCESS macro analysis (Model 3) was used to further investigate the moderating effects of self-enhancement and dominant self-construal [33].

This three-way interaction can be interpreted in two ways: either dominant self-construal moderates the moderating effect of self-enhancement values, or self-enhancement values moderate the moderating effect of dominant self-construal. If self-enhancement values are the only significant first-order moderator, the three-way interaction must be interpreted as a moderated moderation model, where the moderating role of self-enhancement values is further moderated by dominant self-construal [34].

## 3. Results

### 3.1. Descriptive Statistics and Correlation Analysis

As shown by the results of the correlation analysis (Table 1), self-transcendence values were significantly negatively correlated with anxiety levels (*p* < 0.01). Additionally, dominant self-construal was significantly positively correlated with self-transcendence values and significantly negatively correlated with self-enhancement values. Furthermore, self-transcendence values were significantly positively correlated with self-enhancement values.

### 3.2. Analysis of Moderated Moderation

First, we employed a three-step procedure to test whether the relationship between self-transcendence values and anxiety was moderated by dominant self-construal or self-enhancement values. To better understand the interaction effects, participants were categorized based on their levels of self-enhancement values. The interaction term “self-transcendence values × self-enhancement values” was significant (*β* = 0.06, *p* < 0.05) (see Table 2, Model 1-2), whereas the interaction term “self-transcendence values × dominant self-construal” was not significant (*β* = −0.03, *p* > 0.05), indicating that self-enhancement values moderated the relationship between self-transcendence values and anxiety. (The results of testing the moderating role of dominant self-construal are not presented in Table 2). To fully explore the interaction effect between self-transcendence values and self-enhancement values, participants were divided into high (*M* + 1*SD*) and low (*M* − 1*SD*) self-enhancement groups, and simple slope analysis was used to test the moderating role of self-enhancement values (see Figure 2). These results confirm Hypothesis 1, as shown in Figure 2, where self-transcendence values were negatively correlated with anxiety at high levels of self-enhancement values (*β* = −0.15, *p* < 0.001), but the relationship was not significant at low levels of self-enhancement values (*β* = −0.06, *p* > 0.05).

### 3.3. Analysis of the Moderated Moderation Effect of Dominant Self-Construal

After identifying the moderating function of self-enhancement values in the relationship between self-transcendence values and anxiety, we tested Hypothesis 2 using a four-step procedure (see Table 2). The results in Table 2 indicate a significant three-way interaction (*β* = −0.055, *p* < 0.05) involving dominant self-construal, self-transcendence, and self-enhancement values. This interaction suggests that dominant self-construal diminished the moderating influence of self-enhancement values in the relationship between self-transcendence and anxiety, as only self-enhancement values acted as a first-order moderator. To fully explore the three-way interaction, participants were divided into high (*M* + 1*SD*) and low (*M* − 1*SD*) dominant self-construal groups, and simple slope analysis was conducted to test the moderated moderation role of dominant self-construal (see Figure 3). The results show that when an individual’s dominant self-construal was biased toward independence (*M* − 1*SD*) (see Figure 3a), the relationship between self-transcendence values and anxiety was significant only at low levels of self-enhancement values (*β* = −0.198, *p* < 0.001) but not significant at high levels of self-enhancement values (*β* = −0.041, *p* > 0.05). However, when an individual’s dominant self-construal was skewed toward interdependence (*M* + 1*SD*) (see Figure 3b), self-transcendence values significantly and negatively predicted anxiety levels, regardless of the level of self-enhancement values (*β*_low self-enhancement_ = −0.178, *p* < 0.001; *β*_high self-enhancement_ = −0.17, *p* < 0.001) (see Table 3). These results support Hypothesis 2.

## 4. Discussion

Our findings indicate that the interaction between self-enhancement values and self-transcendence values is influenced by self-construal styles. These results highlight the importance of considering cultural and individual differences in the design of public health interventions aimed at reducing anxiety among university students. Specifically, the moderating effect of self-enhancement values was evident when the dominant self-construal was independent rather than interdependent. However, given the limitations of our sample and the cross-sectional design, these findings should be interpreted with caution, and further research is needed to confirm these effects in more diverse populations and longitudinal studies.

Schwartz’s [1] value theory posits that self-transcendence and self-enhancement values are conflicting types. The conflicting nature of these two values has been observed in several empirical studies. In this study, our results support Hypothesis 1 by demonstrating a strong correlation between self-transcendence values and reduced anxiety levels. However, we also found an unexpected strong correlation between self-transcendence and self-enhancement values. This finding may be influenced by the specific cultural and social context of Chinese students, and future studies should aim to replicate these results across different cultural settings. Similar evidence of the simultaneous pursuit of conflicting values has been observed in other studies on Chinese college students’ values. For example, Li [35] found that the values of Chinese college students were characterized by the coexistence of self-transcendence and self-enhancement orientations. Xie et al. [7] also found that some Chinese college students reported high scores across all value types. This may be related to the context of modern Chinese society, where large-scale social changes have led to an increasing focus on the realization of personal goals [36,37]. Simultaneously, socially oriented values are still encouraged, leading to the pursuit of both types of values.

Consistent with Hypothesis 2, the high correlation between self-transcendence and self-enhancement values does not imply that they will have a complementary effect on an individual’s mental health. Our findings reveal that the mitigating effect of self-transcendence values on anxiety was only evident when individuals had low levels of self-enhancement values. In other words, when individuals pursue both types of values, the relationship between self-transcendence and anxiety is disturbed and limited by high levels of self-enhancement values. There are two possible explanations for this. One is that self-enhancement is an induced self-directed value, which itself promotes negative emotions (e.g., anxiety), thereby neutralizing the effect of self-transcendence values on college students’ anxiety and diminishing the mitigating effect. Alternatively, prior studies have shown that individuals who identify with or pursue both self-transcendence values (e.g., traditional Confucian or ethical values) and self-enhancement values (e.g., materialism or Machiavellianism) may experience higher levels of tension and anxiety due to the conflicting nature of these two value types [38,39]. Therefore, the buffering role of self-enhancement values in the relationship between self-transcendence and anxiety observed in the present study may be due to the contradictions and conflicts that arise within Chinese college students who pursue both values simultaneously.

Our findings further support Hypothesis 3, indicating that the moderating effect of self-enhancement values is itself moderated by dominant self-construal. Specifically, for individuals with interdependent dominant self-construal, self-transcendence values consistently had a significant effect on reducing anxiety levels regardless of the level of self-enhancement values. In contrast, for individuals with independent dominant self-construal, self-transcendence values significantly reduced anxiety levels only when self-enhancement values were low. This suggests that the mitigating effect of self-transcendence values on anxiety levels in individuals with interdependent dominant self-construal was not influenced by self-enhancement values. As previously discussed, the interaction between self-transcendence and self-enhancement values plays a critical role in shaping the relationship between self-transcendence values and anxiety. A possible explanation is that individuals with interdependent dominant self-construal tend to espouse collectivism and engage in a high level of dialectical thinking, which makes them less rigid in their self-concept and more tolerant of value conflicts [23]. Consequently, individuals who favor interdependent self-construal may better manage and remain unaffected by the contradictions between self-transcendence and self-enhancement values due to their capacity for dialectical thinking. Jin et al. [40] introduced the concept of “dialectical focus” to describe the integration of individual and modern value orientations with traditional and social values in Chinese culture, rooted in the Confucian doctrine of the Mean. However, the cross-sectional design of this study limits our ability to establish causal relationships, and future longitudinal studies are necessary to explore how these interactions unfold over time.

Although this study presents important findings, it has several limitations that warrant consideration and suggest potential directions for future research. First, this study employed a cross-sectional analytical approach to test the proposed model, which limits our ability to establish causal relationships between the study variables. While it is acknowledged that anxiety may occasionally serve a motivational function under specific conditions (as described by the Yerkes–Dodson Law), this effect is typically limited to moderate and acute forms of anxiety. Numerous studies highlight the detrimental impact of chronic anxiety on mental health when left unmitigated. For instance, research by Kessler et al. [2] and Spinhoven et al. [29] illustrates that addressing chronic anxiety can help prevent the development of comorbid psychological disorders, emphasizing the importance of timely anxiety interventions in mental health strategies. Nevertheless, although significant correlations were found, the inability to infer causality remains a critical limitation. Future longitudinal studies are needed to track how self-transcendence and self-enhancement values evolve over time and how these changes affect anxiety levels. Such designs would allow for a more comprehensive understanding of the temporal dynamics between values and mental health. Second, this study relied on self-reported measures for data collection, which raises the possibility of response biases such as social desirability bias or inaccurate self-assessment. This reliance on self-reporting may lead participants to underreport or exaggerate their anxiety levels, potentially distorting the true relationship between values and mental health. Future research should include more varied data collection methods, such as qualitative interviews or observational assessments, to provide a more nuanced understanding of the participants’ experiences and reduce response bias. Integrating these methods could enrich the findings and offer deeper insights into how values influence anxiety. Third, the sample for this study consisted solely of undergraduate students from public universities in urban areas of China. While this sample offers insights into the experiences of young adults in urban settings, it limits the generalizability of the findings. Future research should include participants from a broader range of contexts, such as students from rural areas or those with different cultural backgrounds, to improve the representativeness of the findings [41]. This would help to determine whether the observed relationships hold across different demographic and cultural groups. Furthermore, expanding the sample to international populations could provide a broader understanding of how self-transcendence and self-enhancement values interact with anxiety across various societal contexts [42].

Additionally, although interaction analyses were conducted, future studies should explore potential confounding variables and include mediation analyses to better understand the underlying mechanisms. Such analyses could clarify how self-transcendence and self-enhancement values affect anxiety and highlight the indirect effects of these value systems. Despite these limitations, this study identified self-enhancement values as a first-order moderator and dominant self-construal as a second-order moderator in the relationship between self-transcendence values and anxiety in Chinese college students. These findings have theoretical implications, showing that self-enhancement can moderate the anxiety-reducing effect of self-transcendence values, challenging the traditional view of these value types as entirely antagonistic. Dominant self-construal further delineates the boundaries of these interactions, suggesting that individuals with interdependent self-construal may better tolerate conflicts between value systems, especially within collectivist cultures. This cultural consideration underscores the significance of understanding value-based influences on mental health. Practically, these findings have practical implications for designing public health interventions specifically for Chinese college students, aimed at reducing anxiety by promoting self-transcendence values tailored to individual differences in self-enhancement tendencies. Given the unique cultural and societal context of Chinese students balancing traditional collectivist values with modern individualistic aspirations, these insights can inform interventions that are culturally sensitive and tailored to this population’s needs. For instance, fostering community involvement or altruistic activities may reduce anxiety, especially in students with high self-enhancement values. Cross-cultural studies examining how value systems interact with mental health can validate these findings and guide mental health interventions across diverse contexts. Moreover, it is valuable to consider the potential bidirectional relationship between trait anxiety and value orientations. Individuals with high trait anxiety may adopt self-transcendence values as a coping mechanism, seeking interpersonal connections to manage anxiety. This suggests that interventions could benefit from fostering adaptive value orientations alongside anxiety reduction, particularly for those predisposed to higher anxiety levels. Future studies should examine whether elevated trait anxiety influences the adoption of self-transcendence values and how this relationship evolves over time.

## Figures and Tables

**Figure 1 behavsci-14-01105-f001:**
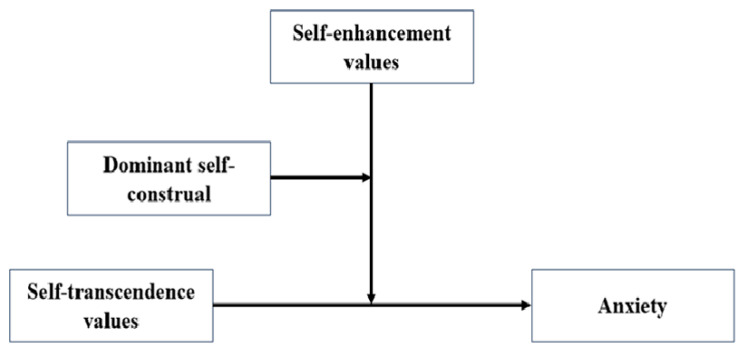
Conceptual model.

**Figure 2 behavsci-14-01105-f002:**
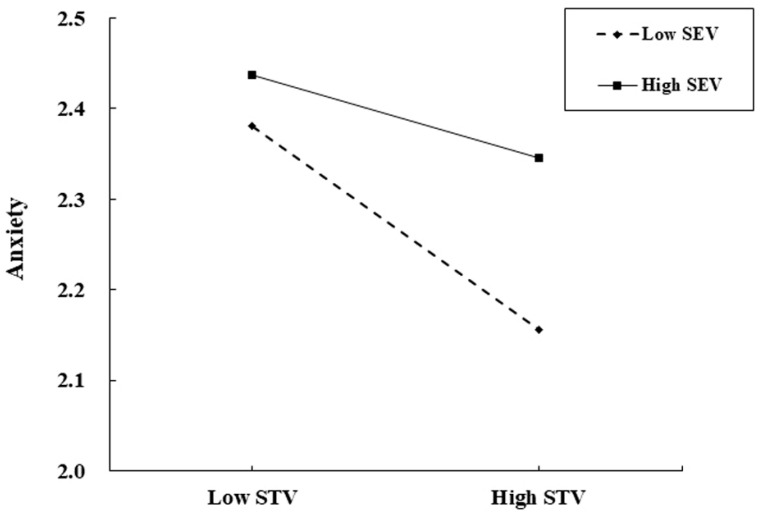
The relationship between self-transcendence values and anxiety across high and low levels of self-enhancement values.

**Figure 3 behavsci-14-01105-f003:**
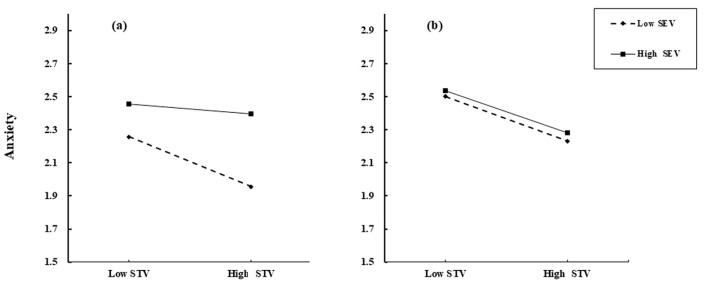
Graph of the moderation effects of self-enhancement values and dominant self-construal on self-transcendence values and anxiety. (**a**) Independent dominant self-construal group; (**b**) Interdependent dominant self-construal group.

**Table 1 behavsci-14-01105-t001:** Means, standard deviations (*SD*), and correlations among variables.

Variable	*M* ± *SD*	STV	SEV	DSC	Anxiety
STV	4.68 ± 0.76	-			
SEV	4.46 ± 0.79	0.46 **	-		
DSC	0.23 ± 0.31	0.18 **	−0.12 **	-	
Anxiety	2.35 ± 0.42	−0.12 **	0.06	0.06	-

Note: STV = self-transcendence values; SEV = self-enhancement values; DSC = dominant self-construal. ** *p* < 0.01.

**Table 2 behavsci-14-01105-t002:** Hierarchical regression analyses.

	Anxiety					
Variables	Model 0	Hypothesis 1		Hypothesis 2		
		Model 1-1	Model 1-2	Model 2-1	Model 2-2	Model 2-3
Gender	−0.02	−0.02	−0.02	−0.02	−0.02	−0.02
Age	−0.05	−0.04	−0.04	−0.03	−0.03	−0.03
STV		−0.11 ***	−0.10 ***	−0.13 ***	−0.13 ***	−0.15 ***
SEV		−0.07 **	0.08 **	0.09 ***	0.10 ***	0.12 ***
DSC				0.06 **	0.06 **	0.07 ***
STV × SEV			0.06 *		0.06 *	0.05 *
STV × DSC					−0.05 *	−0.03
SEV × DSC					−0.09 ***	−0.10 ***
SEV × STV × DSC						−0.06 *
*R* ^2^	0.01	0.04 ***	0.05 ***	0.05 ***	0.09 ***	0.10 ***
Δ*R*^2^		0.03 ***	0.01 *	0.04 ***	0.04 ***	0.01 *

Note: * *p* < 0.05, ** *p* < 0.01, *** *p* < 0.001.

**Table 3 behavsci-14-01105-t003:** Conditional effect of self-transcendence on anxiety at different levels of moderators.

SEV Level	DSC Group	Effect on Anxiety	*SE*	t	95% CI	
					Lower	Upper
Low SEV	Ind DSC	−0.198	0.044	−4.495 ***	−0.284	−0.111
Low SEV	Inter DSC	−0.178	0.041	−4.337 ***	−0.259	−0.097
High SEV	Ind DSC	−0.041	0.035	−1.181	−0.110	0.027
High SEV	Inter DSC	−0.17	0.049	−3.450 ***	−0.266	−0.073

Note: *** *p* < 0.001. Ind DSC = independent dominant self-construal; Inter DSC = interdependence dominant self-construal.

## Data Availability

The raw data are included in the article, further inquiries can be directed to the corresponding author.

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
