# Peer review of "The Impact of Self-Transcendence on Anxiety Among Chinese College Students: The Moderating Roles of Self-Enhancement and Dominant Self-Construal"

_behavsci, 2024, doi:10.3390/bs14111105_

Round 1
Reviewer 1 Report
Comments and Suggestions for Authors
Reviewer Evaluation Report
Title: It should be more precise and avoid redundancies to clarify which variables have been studied and their relationships.
Abstract: Overall, the abstract and keywords are well-crafted, but with some minor adjustments, their effectiveness and precision could be further improved. The abstract does not fully follow the IMRD format. It would be important to include the methodology used and a brief description of the sample (M) and specify the practical implications of the results (D).
Keywords: They should be indexed in thesauri like ERIC to facilitate the location of their work in case it is published.
Introduction: Although it provides a solid theoretical framework based on Schwartz’s theory of values, it lacks a more global perspective that demonstrates a general mastery of the topic, not just related to Chinese culture, to then substantiate the particular choice made. In some points, the writing becomes dense and too technical. Simplifying these parts could make it more accessible to a broader audience. A transition paragraph connecting all sections of the article would help improve the cohesion of the text. Including examples of empirical works illustrating how self-transcendence and self-enhancement values affect anxiety in real contexts could enrich the introduction and make it more tangible.
Hypotheses: They are clear and well-defined; I would add one specifically studying the relationship between self-transcendence and anxiety.
Method: The authors do not indicate what methodology and experimental design they use. It should be stated and justified.
Participants: Although the sample is large, it is limited to students from urban areas. A more detailed justification of why this specific population is relevant to the study and how it was selected could strengthen this section.
Measurement Instruments: Although they have been validated and seem reliable, their description could be improved with information about the psychometric properties of the original scales and their use in other samples from previous studies. Their theoretical justification is also missing.
Data Collection Procedure: It is not exhaustively described how data collection was carried out to ensure the study’s replication, nor how consent was obtained, anonymity and confidentiality of data were ensured, or how the necessary ethical principles for research with human beings were respected.
Data Analysis: Advanced and appropriate statistical analyses are used, such as hierarchical regression and moderated moderation analysis, demonstrating a robust statistical approach suitable for exploring the complex interactions between variables. The inclusion of simple slope analysis to explore interactions provides a clear visualization of how relationships between variables vary at different levels of the moderators. Although control variables are mentioned, a more detailed discussion on how these variables were handled in the analysis could strengthen the methods section.
Results: The study results are solid and provide valuable information about the relationship between personal values and anxiety. The findings have important implications for designing public health interventions and educational strategies, suggesting that promoting self-transcendence values and considering individual differences in self-enhancement and self-construction can help reduce anxiety among university students. However, future research could address some limitations to further strengthen the validity and applicability of the findings. This acknowledgment by the authors is commendable and crucial for transparency and guiding future research.
Discussion: Although the limitations are mentioned, a more detailed discussion on how these limitations might have affected the results and how they could be mitigated in future studies would strengthen this section.
Study Limitations and Suggestions for Future Research: The study cites some limitations, to which I add other suggestions, expandable by the authors if they have the possibility of deeper reflection. The study is cross-sectional, which limits the ability to establish causal relationships between variables. Future longitudinal studies could provide a more comprehensive view of causal relationships. The reliance on self-reported measures may introduce response biases. Including more varied data collection methods, such as qualitative interviews or observational assessments, could enrich the findings and provide a deeper and more nuanced understanding of participants’ experiences. The sample is not representative of all student populations. Including participants from rural areas or different cultural contexts could enrich and improve the representativeness of the findings. Although interaction analyses were conducted, a more detailed exploration of potential confounding variables and the inclusion of mediation analyses could provide a more comprehensive understanding of the underlying mechanisms of the observed relationships.
References: In general, the study’s references are well-selected and provide adequate support for the study’s arguments and findings. However, although most references are recent, some key citations could be updated and supplemented with more recent studies to reflect the latest advances in the field and in other more global cultural contexts as mentioned. Including more international studies could enrich the study’s perspective and provide a broader comparison of how values influence anxiety in different cultural contexts.
Decision: For all the reasons stated, I recommend accepting it with revisions.
Author Response
Comments 1:
Title: It should be more precise and avoid redundancies to clarify which variables have been studied and their relationships.
Response 1:
Thank you for your valuable feedback. We have revised the title based on your suggestion to make it more precise and avoid redundancies.
The original title:Self-Transcendence and Anxiety Among College Students: The Moderating Role of Self-Enhancement and the Moderated Moderating Role of Dominant Self-Construal has been simplified to:
The Impact of Self-Transcendence on Anxiety in College Students: The Moderating Roles of Self-Enhancement and Dominant Self-Construal
This revision more clearly reflects the variables and their relationships while maintaining clarity and focus. We believe the new title better captures the essence of the study.
Thank you again for your constructive input.
Comments 2:
Abstract: Overall, the abstract and keywords are well-crafted, but with some minor adjustments, their effectiveness and precision could be further improved. The abstract does not fully follow the IMRD format. It would be important to include the methodology used and a brief description of the sample (M) and specify the practical implications of the results (D).
Response 2:
Thank you for your helpful feedback. We have revised the abstract to follow the IMRD format more closely. Specifically, we have included a brief description of the methodology and the sample size. The revised abstract now reads:
“This study explores the role of self-transcendence values in reducing anxiety among Chinese undergraduates, with a particular focus on how self-enhancement values and self-construal styles moderate this relationship. A total of 567 undergraduate students were assessed using the Portrait Values Questionnaire (PVQ-21), the Trait Anxiety Inventory (T-AI), and the Self-Construal Scale (SCS). Hierarchical regression and moderated moderation analyses were conducted to test the interactions between the variables. The findings suggest that self-enhancement moderated the association between self-transcendence and anxiety, particularly in individuals with low self-enhancement. Furthermore, the moderating effect of self-enhancement was influenced by dominant self-construal. These findings have practical implications for integrating value-based interventions in educational and public health strategies aimed at improving mental well-being among university students."
We have also clarified the practical implications of the findings for mental health interventions and educational strategies. We believe these changes have improved the precision and clarity of the abstract.
Thank you again for your insightful comments.
Comments 3:
Keywords: They should be indexed in thesauri like ERIC to facilitate the location of their work in case it is published.
Response 3:
Thank you for your suggestion regarding the keywords. We have revised the keywords to ensure they are more suitable for indexing in thesauri such as ERIC. The revised keywords are: “Self-Transcendence”, “Self-Enhancement”, “Self-Construal” ,“Anxiety”, “Mental Health”,“Moderated Moderation”.
Thank you again for your helpful feedback.
Comments 4:
Introduction: Although it provides a solid theoretical framework based on Schwartz’s theory of values, it lacks a more global perspective that demonstrates a general mastery of the topic, not just related to Chinese culture, to then substantiate the particular choice made. In some points, the writing becomes dense and too technical. Simplifying these parts could make it more accessible to a broader audience. A transition paragraph connecting all sections of the article would help improve the cohesion of the text. Including examples of empirical works illustrating how self-transcendence and self-enhancement values affect anxiety in real contexts could enrich the introduction and make it more tangible.
Response 4:
Thank you for your insightful comments on the Introduction. We have made the following revisions based on your suggestions:
- We have expanded the discussion to include research and examples from various cultural contexts beyond China. This now includes references to studies from Western cultures, such as the United States and Europe, as well as other Eastern cultures, including South Korea and Japan. These additions provide a broader, global perspective on how self-transcendence and self-enhancement values impact mental health.
- We have simplified some of the dense and technical sections in the introduction to make the writing more accessible to a wider audience. This ensures that both specialists and non-specialists can follow the theoretical discussion more easily.
- We have incorporated additional empirical examples to illustrate how self-transcendence and self-enhancement values influence anxiety in real-life contexts. These examples help to make the introduction more tangible and grounded in practical findings.
- A transition paragraph has been added to improve the cohesion of the text, linking the theoretical framework to the following sections of the manuscript.
Comments 5:
Method: The authors do not indicate what methodology and experimental design they use. It should be stated and justified.
Response 5:
Thank you for your feedback regarding the methodology. We have revised the Methods section to explicitly state the research design and justify its use. The revised section now reads:
“This study employed a cross-sectional survey design to explore the relationship between self-transcendence, self-enhancement, and anxiety among Chinese college students... The use of a cross-sectional design was deemed appropriate for this study as it allows for the efficient collection of data from a large sample to examine correlations between variables at a single point in time. Although causal relationships cannot be established through cross-sectional studies, this design provides valuable insights into potential areas for future longitudinal research.”
We believe this revision addresses your concern about the lack of clarity regarding the methodology and its justification.
Thank you again for your valuable feedback.
Comments 6:
Participants: Although the sample is large, it is limited to students from urban areas. A more detailed justification of why this specific population is relevant to the study and how it was selected could strengthen this section.
Response 6:
Thank you for your feedback regarding the Participants section. We have revised the text to provide a more detailed justification for selecting students from urban areas. The revised section now reads:
“The choice of urban university students was intentional for two key reasons. First, urban areas in China have been experiencing rapid modernization, leading to unique pressures on students, such as high academic expectations and career demands. These pressures are particularly relevant when studying anxiety levels and value-based behavior in educational settings. Second, students in urban areas are more likely to be influenced by both traditional collectivist values and emerging modern individualistic values, making them an ideal group to investigate the interaction between self-transcendence and self-enhancement in relation to anxiety.”
We believe this additional explanation clarifies the relevance of this specific population to the study.
Thank you again for your valuable feedback.
Comments 7:
Measurement Instruments: Although they have been validated and seem reliable, their description could be improved with information about the psychometric properties of the original scales and their use in other samples from previous studies. Their theoretical justification is also missing.
Response 7:
Thank you for your feedback regarding the measurement instruments. We have revised the section to provide more detailed information about the psychometric properties of the scales and their use in previous studies. The revised section now includes:
Portrait Values Questionnaire (PVQ-21): We have provided details on its internal consistency (Cronbach's α values ranging from 0.70 to 0.80) and validated use in Chinese and international samples [18, 19].
Trait Anxiety Inventory (T-AI): The T-AI was originally developed by Spielberger et al. in 1970 and later adapted to Chinese in 1995 [31]. We provided details on its internal consistency (Cronbach’s α = 0.91 for trait anxiety and 0.88 for state anxiety) and its use in previous Chinese studies. In this study, the Cronbach’s alpha for the T-AI was 0.85, indicating good internal consistency.
Self-Construal Scale (SCS): We have expanded the description to include the details of the scale, noting that it consists of 24 items (12 items for each subscale) and uses a 7-point Likert scale. We also included the Cronbach's alpha values for interdependent (0.88) and independent (0.75) self-construal, as validated in Chinese samples [21].
We have also included a theoretical justification for the use of these specific scales, emphasizing their relevance to the constructs examined in this study.
Thank you again for your valuable feedback.
Comments 8:
Data Collection Procedure: It is not exhaustively described how data collection was carried out to ensure the study’s replication, nor how consent was obtained, anonymity and confidentiality of data were ensured, or how the necessary ethical principles for research with human beings were respected.
Response 8:
Thank you for your insightful comments regarding the data collection procedure. We have revised the manuscript to provide a more detailed explanation of how the data were collected, how consent was obtained, and how participant anonymity and confidentiality were ensured. The updated text now reads:
“Participants were recruited between September and November 2023, and data were collected in a classroom setting with the permission of the university administration. Students were informed about the purpose of the study, the procedures involved, and the voluntary nature of participation, and they provided written informed consent. It was emphasized that they could withdraw from the study at any time without any consequences. To protect privacy, each questionnaire was assigned a unique identification code, and participants were not required to provide any personally identifiable information. All data were securely stored and were accessible only to the research team for analysis. This study adhered to the ethical guidelines of the Declaration of Helsinki and was approved by the Ethics Committee of the Faculty of Psychology at Southwest University (Approval No. H24040).”
This revision clarifies the data collection procedure, ensuring transparency and ethical compliance in handling participant information.
Thank you again for your valuable feedback.
Comments 9:
Data Analysis: Advanced and appropriate statistical analyses are used, such as hierarchical regression and moderated moderation analysis, demonstrating a robust statistical approach suitable for exploring the complex interactions between variables. The inclusion of simple slope analysis to explore interactions provides a clear visualization of how relationships between variables vary at different levels of the moderators. Although control variables are mentioned, a more detailed discussion on how these variables were handled in the analysis could strengthen the methods section.
Response 9:
Thank you for your helpful feedback on the data analysis section. We have revised the manuscript to provide a more detailed explanation of how control variables were handled in the analysis. The revised section now reads:
“Control variables, such as age and gender, were included in Model 0 to account for their potential confounding effects on anxiety. By introducing these demographic factors in the first step of the hierarchical regression, we ensured that their influence was controlled before analyzing the main effects and interactions.”
We believe this revision clarifies the handling of control variables and strengthens the methods section by addressing your concerns.
Thank you again for your valuable feedback.
Comments 10:
Results: The study results are solid and provide valuable information about the relationship between personal values and anxiety. The findings have important implications for designing public health interventions and educational strategies, suggesting that promoting self-transcendence values and considering individual differences in self-enhancement and self-construction can help reduce anxiety among university students. However, future research could address some limitations to further strengthen the validity and applicability of the findings. This acknowledgment by the authors is commendable and crucial for transparency and guiding future research.
Response 10:
Thank you for your positive feedback on the results section and your valuable suggestions for future research. We have revised the manuscript to provide a more detailed discussion of the study's limitations and have added additional suggestions for future research. Specifically:
We have highlighted the need for future research to explore these relationships using longitudinal designs to establish causal links between personal values and anxiety.
We have discussed the potential impact of cultural and contextual factors on the generalizability of our findings, suggesting that cross-cultural studies could further validate the applicability of promoting self-transcendence values to reduce anxiety in diverse populations.
We believe these revisions address your comments and enhance the clarity and rigor of the manuscript. Thank you again for your valuable feedback.
Comments 11:
Discussion: Although the limitations are mentioned, a more detailed discussion on how these limitations might have affected the results and how they could be mitigated in future studies would strengthen this section.
Response 11:
Thank you for your insightful feedback regarding the need for a more detailed discussion on how the study's limitations might have influenced the results and how they could be mitigated in future research. We have revised the discussion section to address these concerns. Specifically, we have expanded our discussion of the cross-sectional design and the limitations it imposes on establishing causal relationships. We have also acknowledged the reliance on self-reported measures, which could introduce response biases. Future research directions now include the use of longitudinal designs to establish causal relationships and the incorporation of qualitative methods, such as interviews and behavioral observations, to provide a more nuanced understanding of the relationships between values and anxiety. Additionally, we have discussed the limitations of our sample, which consisted solely of Chinese urban university students, and recommended that future studies include more diverse populations to improve the generalizability of our findings. These revisions provide a clearer understanding of the study's limitations and offer concrete suggestions for mitigating them in future research.
Comments 12:
Study Limitations and Suggestions for Future Research: The study cites some limitations, to which I add other suggestions, expandable by the authors if they have the possibility of deeper reflection. The study is cross-sectional, which limits the ability to establish causal relationships between variables. Future longitudinal studies could provide a more comprehensive view of causal relationships. The reliance on self-reported measures may introduce response biases. Including more varied data collection methods, such as qualitative interviews or observational assessments, could enrich the findings and provide a deeper and more nuanced understanding of participants’ experiences. The sample is not representative of all student populations. Including participants from rural areas or different cultural contexts could enrich and improve the representativeness of the findings. Although interaction analyses were conducted, a more detailed exploration of potential confounding variables and the inclusion of mediation analyses could provide a more comprehensive understanding of the underlying mechanisms of the observed relationships.
Response 12:
Study Limitations and Suggestions for Future Research: The study cites some limitations, to which I add other suggestions, expandable by the authors if they have the possibility of deeper reflection. The study is cross-sectional, which limits the ability to establish causal relationships between variables. Future longitudinal studies could provide a more comprehensive view of causal relationships. The reliance on self-reported measures may introduce response biases. Including more varied data collection methods, such as qualitative interviews or observational assessments, could enrich the findings and provide a deeper and more nuanced understanding of participants’ experiences. The sample is not representative of all student populations. Including participants from rural areas or different cultural contexts could enrich and improve the representativeness of the findings. Although interaction analyses were conducted, a more detailed exploration of potential confounding variables and the inclusion of mediation analyses could provide a more comprehensive understanding of the underlying mechanisms of the observed relationships.
Comments 13:
References: In general, the study’s references are well-selected and provide adequate support for the study’s arguments and findings. However, although most references are recent, some key citations could be updated and supplemented with more recent studies to reflect the latest advances in the field and in other more global cultural contexts as mentioned. Including more international studies could enrich the study’s perspective and provide a broader comparison of how values influence anxiety in different cultural contexts.
Response 13:
Thank you for your valuable feedback on our reference list. We appreciate the suggestion to update and expand the references to include more recent studies and further integrate global perspectives. Based on your comments, we have reviewed our references and made targeted additions to address recent advances in the field and provide a broader, more global perspective on the relationship between values and mental health.
1.We added recent studies that address value-related mental health impacts across diverse cultural contexts, such as Ardenghi et al. (2023) and Xie et al. (2023), which analyze values among university students in distinct cultural settings. These studies provide updated insights into the relationship between values, empathy, and anxiety across different cultural frameworks.
2.To further enrich our understanding of international value structures and anxiety, we included studies that compare value systems across East Asian and Western cultural contexts, such as Markus & Kitayama (1991) and Rudnev et al. (2018). These references allow for a more nuanced exploration of cultural distinctions in self-construal and value conflicts.
3.We integrated additional international sources, including studies on adolescents and adult populations from countries outside China, such as Watanabe et al. (2020) and Hanel et al. (2020). These papers extend the cultural scope and provide additional support for cross-cultural comparisons.
We believe these revisions strengthen the foundation of our study and provide a more comprehensive view of how values impact mental health across diverse sociocultural backgrounds.
Comments 14:
Decision: For all the reasons stated, I recommend accepting it with revisions.
Response 14:
Thank you very much for your thorough review and support for our study. We have carefully reviewed and addressed each of your recommended revisions to enhance the clarity and scholarly quality of the paper. All revisions have been highlighted in the updated manuscript, and we hope these changes meet your expectations.
We are grateful for your valuable suggestions, which have helped to improve the quality of our work. We look forward to any further feedback you may have!
Reviewer 2 Report
Comments and Suggestions for Authors
The paper Self-Transcendence and Anxiety Among College Students: The Moderating Role of Self-Enhancement and the Moderated Moderating Role of Dominant Self-Construal explored variables which may affect anxiety in Chinese college students. The authors concluded that a self-enhancement values orientation may moderate this relationship, further moderated by self-construal. This is then asserted to have potential implications for improving mental health.
Overall, I found the paper very well written. The review of Schwartz’s theory was clear and set the study up nicely. I found the methods section easy to follow and the results quite clear.
My major theoretical concern was the author’s assumption that reducing anxiety would improve mental health. There is certainly literature supporting the notion that anxiety plays a motivational function, such as that predicted by Yerkes-Dodson law. This would seem especially salient for a sample of college students. I could see where reducing their anxiety while in school could lead to more negative mental health outcomes in the future. Thus if the crux of the paper’s significance is that mental health can be improved by anxiety reduction, I recommend more support for this relationship. Along those lines, might trait anxiety be higher in those that are attending college? That is to say, might the sample be biases to comprise individuals high in trait anxiety? Along those lines, I would have liked if the authors had described what level of trait anxiety on the measure indicates “anxiety-related problems”.
Along those lines, while acknowledging that causality is not established, the future directions suggest that the promotion of self-transcendence values could improve mental health and quality of life, yet it seems just as likely that those with trait anxiety are motivated towards embracing self-enhancement values – essentially targeting a side effect rather than a cause.
As one minor suggestion, I found the title very clunky. I think a more streamlined title could convey the overall meaning of the work, with more specifics left to the abstract.
Otherwise, I found the article a nice read and feel that it adds to the literature.
Author Response
Comments 1:
The paper Self-Transcendence and Anxiety Among College Students: The Moderating Role of Self-Enhancement and the Moderated Moderating Role of Dominant Self-Construal explored variables which may affect anxiety in Chinese college students. The authors concluded that a self-enhancement values orientation may moderate this relationship, further moderated by self-construal. This is then asserted to have potential implications for improving mental health.
Response 1:
Thank you for summarizing our findings. We aimed to show that self-enhancement moderates the link between self-transcendence and anxiety, with an additional moderation by self-construal. These results underscore how values and self-construal can shape mental health outcomes in culturally specific ways. We hope our study offers insights for designing targeted mental health interventions in educational settings.
Comments 2:
Overall, I found the paper very well written. The review of Schwartz’s theory was clear and set the study up nicely. I found the methods section easy to follow and the results quite clear.
Response 2:
Thank you very much for your positive feedback on the clarity of our writing, as well as on our review of Schwartz’s theory, the methodology, and the results. We appreciate your encouragement and are glad that our presentation effectively supported the study’s objectives and findings.
Comments 3:
My major theoretical concern was the author’s assumption that reducing anxiety would improve mental health. There is certainly literature supporting the notion that anxiety plays a motivational function, such as that predicted by Yerkes-Dodson law. This would seem especially salient for a sample of college students. I could see where reducing their anxiety while in school could lead to more negative mental health outcomes in the future. Thus if the crux of the paper’s significance is that mental health can be improved by anxiety reduction, I recommend more support for this relationship. Along those lines, might trait anxiety be higher in those that are attending college? That is to say, might the sample be biases to comprise individuals high in trait anxiety? Along those lines, I would have liked if the authors had described what level of trait anxiety on the measure indicates “anxiety-related problems”.
Response 3:
Thank you for your valuable feedback regarding the assumption that reducing anxiety would improve mental health. We have revised the manuscript to address this point in the introduction, methods, and discussion sections.
In the introduction, we clarified that while certain levels of acute anxiety may serve a motivational function, persistent and unmitigated chronic anxiety is linked to adverse mental health outcomes, as supported by relevant research (e.g., Kessler et al., 2005; Spinhoven et al., 2014). This contextualization underscores the importance of managing chronic anxiety to promote mental health, particularly among college students, whose academic pressures often result in elevated levels of sustained anxiety.
To address the reviewer’s question on trait anxiety, we provided additional information in the methods section, explaining that we utilized the Trait Anxiety Inventory to capture participants’ enduring anxiety patterns rather than situational anxiety. This tool’s validation in Chinese college samples allows us to reliably assess chronic anxiety and align with the study's focus on anxiety’s longer-term mental health impacts.
Lastly, in the discussion, we have expanded on the implications of managing chronic anxiety within educational contexts, noting that interventions can be designed to distinguish between beneficial, acute anxiety responses and harmful, prolonged anxiety patterns. These distinctions underscore the need for targeted mental health strategies for college students experiencing chronic anxiety, aligning with the study’s focus on sustainable mental health improvement.
Comments 4:
Along those lines, while acknowledging that causality is not established, the future directions suggest that the promotion of self-transcendence values could improve mental health and quality of life, yet it seems just as likely that those with trait anxiety are motivated towards embracing self-enhancement values – essentially targeting a side effect rather than a cause.
Response 4:
We appreciate the reviewer’s insightful suggestion regarding the bidirectional relationship between trait anxiety and value orientations. In response, we have revised the discussion section to clarify the potential for trait anxiety to influence the adoption of self-transcendence values. We acknowledge that individuals with elevated trait anxiety may be inclined to adopt these values as a coping mechanism, potentially targeting anxiety’s symptoms rather than its root causes. To address this, we have added a consideration that future interventions could benefit from fostering adaptive value orientations in tandem with anxiety reduction efforts. This revised discussion also emphasizes the need for further research on the temporal dynamics between trait anxiety and value adoption to clarify these complex relationships.
Comments 5:
As one minor suggestion, I found the title very clunky. I think a more streamlined title could convey the overall meaning of the work, with more specifics left to the abstract.
Response 5:
We appreciate the reviewer’s suggestion regarding the title's clarity and conciseness. Based on this feedback, we have streamlined the title to better capture the study’s focus while avoiding excessive detail. The revised title, "The Impact of Self-Transcendence on Anxiety in College Students: The Moderating Roles of Self-Enhancement and Dominant Self-Construal," provides a clearer summary of the study’s key elements and aligns with the suggestion to leave specific details to the abstract. Thank you for this helpful recommendation.
Comments 6:
Otherwise, I found the article a nice read and feel that it adds to the literature.
Response 6:
We sincerely appreciate the reviewer’s positive feedback and are pleased to hear that our article adds value to the literature. Thank you for recognizing our efforts, and we hope that the revisions have further strengthened the manuscript.
Round 2
Reviewer 1 Report
Comments and Suggestions for Authors
Reviewer’s Response to the Authors’ Changes:
Response 1: The authors have satisfactorily addressed the suggested change for the title.
Response 2: The authors have satisfactorily addressed the suggested change for the abstract.
Response 3: The authors have satisfactorily addressed the suggested change for the keywords.
Response 4: The authors have satisfactorily addressed the suggested change for the introduction and have improved the theoretical foundation of their work.
Response 5: The authors have specified that their study employed a cross-sectional survey design. There is confusion between the concepts of design and method. The authors should first specify which method they are using and then clarify the type of design, justifying this designation with a reference from an author specialised in research methodology. This aspect is important and needs to be addressed. I am adding a reference that might be useful: Creswell, J. W., & Creswell, J. D. (2017). Research Design: Qualitative, Quantitative, and Mixed Methods Approaches (5th ed.). SAGE Publications.
Response 6: The authors have satisfactorily addressed the suggested change for the participants. They might consider including in the title of their work and in the practical implications of their results that they are directed at this population. Otherwise, the reader might view their sample as biased and unrepresentative, which would diminish the value of the work’s contributions.
Response 7: The authors have satisfactorily addressed the suggested change for the study instruments.
Response 8: The authors have satisfactorily addressed the suggested change for the data collection procedure section.
Response 9: The authors have satisfactorily addressed the suggested change for the data analysis section.
Responses 10-12: The authors have satisfactorily addressed the suggested changes for the discussion, conclusions, limitations, future directions, and practical implications of the study.
Response 13: The authors have satisfactorily addressed the suggested change for the references section.
Response 14: The authors have satisfactorily addressed most of the suggested changes. Only two aspects of their work remain to be reviewed: firstly, revising the possibility of a change in the title and in the practical implications of the results, considering the comment made in this evaluation report for the participants section; and secondly, the methodology section, to resolve the existing confusion between method and design, and to clarify the method and design of the research employed in a well-founded manner. Once these two aspects are addressed, I would approve it for publication.
Author Response
Comment 1
The authors have satisfactorily addressed the suggested change for the title.
Response:
Thank you for confirming the title adjustment. We have ensured that the title now clearly reflects the focus on Chinese college students, which aligns with the reviewer’s guidance to improve clarity and context.
Comment 2
The authors have satisfactorily addressed the suggested change for the abstract.
Response:
We appreciate the reviewer’s confirmation on the abstract. The revision was intended to provide a clearer summary of our study's objectives, methodology, and findings, with emphasis on the study’s specific focus and contributions.
Comment 3
The authors have satisfactorily addressed the suggested change for the keywords.
Response:
Thank you for the feedback. We have refined the keywords to enhance discoverability and better align with the core themes of the study.
Comment 4
The authors have satisfactorily addressed the suggested change for the introduction and have improved the theoretical foundation of their work.
Response:
We are grateful for the reviewer’s positive feedback. Our revisions focused on strengthening the theoretical framework, particularly regarding Schwartz’s value theory and its application within different cultural contexts, to establish a strong foundation for the study.
Comment 5
The authors have specified that their study employed a cross-sectional survey design. There is confusion between the concepts of design and method. The authors should first specify which method they are using and then clarify the type of design, justifying this designation with a reference from an author specialised in research methodology. This aspect is important and needs to be addressed. I am adding a reference that might be useful: Creswell, J. W., & Creswell, J. D. (2017). Research Design: Qualitative, Quantitative, and Mixed Methods Approaches (5th ed.). SAGE Publications.
Response:
Thank you for pointing out the need for clarity between the method and design. We have revised the methodology section to specify the quantitative survey method and the cross-sectional design used in this study. Additionally, we included a reference to Creswell and Creswell (2017) to substantiate our methodological approach, as recommended.
Comment 6
The authors have satisfactorily addressed the suggested change for the participants. They might consider including in the title of their work and in the practical implications of their results that they are directed at this population. Otherwise, the reader might view their sample as biased and unrepresentative, which would diminish the value of the work’s contributions.
Response:
We appreciate this feedback. The title has been updated to specify the target population as Chinese college students, thus providing clarity on the sample used. Additionally, we revised the practical implications section in the discussion to emphasize that the study’s findings are specifically relevant to Chinese college students, underscoring the cultural and demographic context.
Comment 7
The authors have satisfactorily addressed the suggested change for the study instruments.
Response:
Thank you for your confirmation. We ensured that each instrument’s relevance and validity for measuring key constructs in this study were thoroughly addressed.
Comment 8
The authors have satisfactorily addressed the suggested change for the data collection procedure section.
Response:
We appreciate the positive feedback on this section. The data collection process was clarified to ensure transparency regarding participant recruitment and informed consent procedures.
Comment 9
The authors have satisfactorily addressed the suggested change for the data analysis section
Response:
Thank you for your feedback. We have detailed our approach to data analysis, including hierarchical regression analysis and the use of PROCESS macro for testing moderation and moderated moderation effects.
Comment 10-12
The authors have satisfactorily addressed the suggested changes for the discussion, conclusions, limitations, future directions, and practical implications of the study.
Response:
We are pleased that these sections meet the reviewer’s expectations. In the discussion and conclusions, we have aimed to contextualize the findings in light of the study’s limitations and suggest actionable insights for mental health interventions within educational settings, especially tailored for Chinese college students.
Comment 13
The authors have satisfactorily addressed the suggested change for the references section.
Response:
Thank you. We ensured that all references are accurate, up-to-date, and relevant to support the arguments and findings presented in the manuscript.
Comment 14
The authors have satisfactorily addressed most of the suggested changes. Only two aspects of their work remain to be reviewed: firstly, revising the possibility of a change in the title and in the practical implications of the results, considering the comment made in this evaluation report for the participants section; and secondly, the methodology section, to resolve the existing confusion between method and design, and to clarify the method and design of the research employed in a well-founded manner. Once these two aspects are addressed, I would approve it for publication.
Response:
We have made the suggested adjustments as follows:
- Title and Practical Implications: The title now reflects the specific focus on Chinese college students. In the practical implications, we have emphasized the applicability of the findings to this demographic, reducing any potential concerns about sample bias.
- Methodology Section: We clarified the method as a quantitative survey and the design as cross-sectional. To provide a theoretical foundation for these choices, we included a citation to Creswell and Creswell (2017), as recommended.
We hope that these final adjustments address the remaining points, and we thank the reviewer for their detailed guidance throughout this process.